# TESTING FOR OUTLIERS IN A HIDDEN FUNCTION

## ABSTRACT

Testing for outliers is an important data mining task, rooted in learning theory, which aims at discovering points that deviate from the most considered "normal". It is widely applicable to identify intrusion, fraud, anomalies, but also values that may occur rarely but are important for various data analysis applications, e.g., maximum/minimum, median, etc. We consider a deterministic version of the problem, called Testing for Hidden Function's Outliers (HFO-testing for short), defined as follows. Given a hidden function $f$ of at most $\ell$ values, the goal is to find all outliers of $f$, that is, values whose preimages are of size at most $k$, together with their preimages, where $\ell, k$ are the problem parameters. Finding outliers can be done by asking OR queries, each represented by a set of pairs $(x, y)$, where the answer to each query is 1 if at least one pair in the query is consistent with function $f$, i.e., a pair $(x, f(x))$ belongs to the query for some $x$, and 0 otherwise. We formally model this process as a learning game between two players: the adversary, who first chooses&hides a function and later provides feedback to the other player queries, and the other player (user) who creates and asks queries and later analyzes the obtained feedback. This paper aims at finding a short universal sequence of queries that allows the user to solve the above-mentioned problem for any adversarial function $f$ from any given (potentially very large) domain $N$ to a codomain $M$. We formally prove nearly-cubic, in terms of parameters $\ell, k$ and polylog$(N, M)$, upper and lower bounds for this problem, which are tight up to a polylogarithmic factor. The upper bound is showed by constructing and analyzing non-adaptive deterministic OR-query system, with decoding. The lower bound is proved by designing "costly" functions for any given OR-query system.

## 1 INTRODUCTION

The vast majority of machine learning (ML) research focuses on discovering statistically significant events and values. One of the classic problems is a heavy-hitter problem, in which the goal is to identify elements with frequency of occurrence above a given threshold, see e.g., Cormode et al. (2003); Cormode & Muthukrishnan (2005); Cormode & Hadjieleftheriou (2008); Yu et al. (2004); Kowalski & Pajak (2022). Other related problems include quantile tracking Cormode et al. (2005); Gilbert et al. (2002b); Greenwald & Khanna (2001) and approximate histogram maintenance and reconstruction Gibbons et al. (2002); Gilbert et al. (2002a). However, sometimes even (statistically) non-significant values, commonly called *outliers*, could have a huge impact on learning process, by creating an unexpected bias or an information noise. The problems of identifying outliers has been considered in the context of specific scenarios and applications, e.g., anomaly detection or noise cancellation, focused on identifying/mitigating unusual stochastic events, see e.g., Hawkins (1980); Barbará et al. (2006); Krishna et al. (2022); Nedelkoski et al. (2019).

All known testing methods for detecting outliers, considered in the literature, focused on designing and implementing various *statistical tests*, even in case of adversarial setting, c.f., Roth et al. (2019). Inspired by the Combinatorial Group Testing field, see e.g., Du et al. (2000), in this work we abstract and study a generic problem of identifying outliers in a hidden function, using a *deterministic* query system. In each query, also called an OR-query, a set of "guesses" is submitted and a positive feedback emphasizes that at least one of the guesses in correct (but without revealing more information, such as how many guesses are correct or which ones are).[1] We are interested in a universal system of

---

[1]This kind of feedback is among simplest and fundamental feedbacks used in query systems, group testing and information theory, c.f., Du et al. (2000); Kautz & Singleton (1964). An equivalent concept (to systems of

queries that could outlier values, and their inverse images, for any hidden function. One of the visual examples is the well-known Game of Mastermind (see e.g., Knuth (1977); Doerr et al. (2016)): one player would like to learn which colors are rarely used by the opponent, and the only way to do it is by asking questions about colors (corresponding to the values of the function chosen&hidden by the opponent) at specific positions of the opponent's board (corresponding to the arguments of the hidden function). The formal definition of the problem is given in Section 1.1.

The combinatorial group testing (GT), which has been an inspiration for deterministic identification of outliers by using a query system, is a related, yet more simplistic testing framework, focusing on finding elements in a *hidden set*. It has been thoroughly studied since 1943 Dorfman (1943). Group testing, together with closely related superimposed codes, have already found many applications in machine learning, the most prominent include: approximating the nearest neighbor Engels et al. (2021), simplifying multi-label classifiers Ubaru et al. (2020), accelerating forward pass of a deep neural network Liang & Zou (2021), or dimensionality reduction and error correction in online decision making Hajiaghayi et al. (2024).

As we will argue later, applying this framework to the HFO problem, by treating the function as a subset of pairs, is very inefficient (in the sense that the resulting query system is far too long than desired), therefore in this work we propose more complex and efficient approach.

## 1.1 THE PROBLEM OF TESTING FOR HIDDEN FUNCTION'S OUTLIERS (HFO)

We consider the following problem of *Testing for Hidden Function's Outliers* (*HFO problem*, for short). Suppose there are two players, and one of them (also called an *adversary*) chooses a function $f : [N] \to [M]$, for some domain $[N] = \{0, 1, \ldots, N-1\}$ and codomain $[M] = \{0, 1, \ldots, \}$. The other player, also called a *user*, only knows the sizes of domain, $N$, and codomain, $M$. W.l.o.g. and to simplify notation, $f$ could be also viewed as the set of pairs $\{(x, f(x)\}_{x \in [N]}$. Suppose that the image of function $f$ is of size at most $\ell$, i.e., $|f([N])| \leq \ell$, and parameter $\ell \leq M$ (or its upper bound) is also known to the user. For a given parameters $N, M, \ell$ and parameter $k$, an $(N, M, \ell, k)$-HFO problem is to output by the user all pairs $(y, f^{-1}(y))$, over $y \in [M]$, such that $|f^{-1}(y)| \leq k$. Such values $y$ are called $k$-*rare* values (of unknown function $f$), and its value depends on how rare values of the function (chosen in secret by the adversary) the user wants to identify. W.l.o.g. we assume that all parameters are positive integers.

We consider solutions of the HFO problem designed in a *Query System (QS)*. More precisely, in order to solve the HFO problem, the user keeps submitting queries $Q_1, Q_2, \ldots, Q_\tau$, each of them being a subset of $[N] \times [M]$ and $\tau$ being the number of queries in the considered/designed QS. After submitting a query $Q_i$, the user receives a feedback $\Phi_i = \left\lceil \frac{|Q \cap f|}{|Q|} \right\rceil$ from the adversary (who knows function $f$, and therefore can compute and provide such feedback to the user). In other words, $\Phi_i = 1$ if $|Q \cap f| \geq 1$ and $\Phi_i = 0$ otherwise. Intuitively, each query $Q_i$ stores a set of guesses for possible values of the function $f$ for some arguments (it could contain several guesses for the same argument), and if at least one of them is correct, feedback $\Phi_i = 1$ is received (otherwise the feedback is 0). Because of the feedback definition, these types of queries are also called *OR-queries*.

We study *non-adaptive deterministic* query systems, that is, the queries are fixed by the user before starting testing a hidden function. Such queries could be used to finding outliers in *any function* (for given parameters $N, M, \ell, k$), therefore they are also called *universal*. The $k$-rare values of the hidden function and their reverse images need to be computed by the user based on the queries (designed and asked by the user) and the query feedback (received from the adversary, defined earlier), which is a binary vector $\Phi = \langle \Phi(1), \ldots, \Phi(\tau) \rangle$. (Recall that $\tau$ denotes the number of queries in the considered user's query system.)

## 1.2 OUR TECHNICAL CONTRIBUTION

Our main contribution is a deterministic algorithm constructing a QS of length
$$(k + \ell)k\ell \operatorname{polylog}(N, M, \ell, k) ,$$

---

OR-queries and feedback) is computability by OR-NOT logic circuit, in which gates in odd layers are OR-gates and gates in even layers are NOT-gates (such simple logic gates and circuits can be efficiently implemented in majority of ML models). While OR-gates are intuitively relevant to OR-queries, we will discuss relevance of NOT-gates later on when discussing a decoding algorithm.

which solves $(N, M, \ell, k)$-HFO (for any hidden function), see Section 2 and Theorem 1. Here, $\mathrm{polylog}(N, M, \ell, k)$ denotes some small-degree small-constant polylogarithmic function in $N, M, \ell, k$, to be specified in more details in Theorem 1. It is accompanied by a decoding algorithm – both algorithm working in time polynomial in $N, M$ (and thus, also polynomial in $\ell, k$, since $\ell \leq M$ and $k \leq N$). We also show that the constructed QS can be enhanced to tolerate some number of adversarial errors in the feedback vector (i.e., the adversary may not provide honest/correct feedback to some queries). Our QS can be applied to both deterministic and random functions – in the former case, it could help in discovering adversarial anomalies in deterministic systems (e.g., in systems security), in the latter case – to filter out anomalies and biases (especially adversarial) in stochastic learning.

We also prove almost matching (up to a small polylogarithmic factor) lower bound on the length of any QS solving $(N, M, \ell, k)$-HFO (for any hidden function). Specifically, in Theorem 2 in Section 3, we prove that any such QS must have at least $\Omega(\ell \cdot \min\{k^2 \log_k N, N\})$ queries.

### 1.3 WHY GROUP TESTING IS INEFFICIENT FOR HFO

In Group Testing, the goal is to discover a hidden subset of a given universe by asking queries (which are subsets of the universe) and analyzing the feedback. The classic OR feedback is the same as considered in the HFO problem: if the intersection of the hidden set with the query is non-empty then the feedback is 1, otherwise is 0. Assuming that the size of the universe is $D$ and the size of the hidden set at most $d$, it is already known that GT can be solved in this feedback model by using $O(d^2 \log(D/d))$ queries, see e.g., De Bonis et al. (2003), and an explicit polynomial-time construction of length $O(d^2 \log D)$ exists Porat & Rothschild (2011). The best known lower bound on the number of queries is $\Omega(\min\{d^2 \log D / \log d, D\})$ Clementi et al. (2001). [2]

One could attempt applying such framework to the HFO problem, by considering a function as a set of $d = N$ pairs in the universe (of pairs $[N] \times [M]$) of size $D = N \cdot M$. However, the above-mentioned lower bound implies that such approach would result in query systems containing at least $\Omega(\min\{N^2 \cdot \frac{\log(N \cdot M)}{\log N}, N \cdot M\})$ queries Clementi et al. (2001). The number of queries in our construction depends only *polylogarithmically* on $N$ and $M$.

**Paper overview.** Section 2 provides the main technical contribution, which is a construction (and decoding algorithm) of a universal QS solving $(N, M, \ell, k)$-HFO for any hidden function, with fault-tolerant enhancement in Section 2.1. Section 3 provides nearly matching lower bound for the number of queries in query systems solving $(N, M, \ell, k)$-HFO. Additional discussion (to the one placed already in Section 1) of the results and potential future work is deferred to Appendix A, while Appendix B summarizes limitations of our setting.

## 2 UNIVERSAL QS FOR HFO

Suppose that the adversary selected a hidden function $f : [N] \to [M]$ such that $|f([N])| \leq \ell$. Parameters $N, M, \ell$ are known to the user (i.e., designer of the query system) in the beginning, but no other information about function $f$ is known. The user chooses parameter $k - 1$, being the cap on the size of the inverse image of the rare values to select. (Without loss of generality, in this section we deliberately chose to re-scale parameter $k$ and perform the construction, decoding and analysis for $(k - 1)$-rare values, as technical arguments and formulas for $k - 1$ are a bit shorter than for $k$.)

In this section, we design one universal query system (QS) which allows learning all $(k - 1)$-rare values of any function $f$ chosen&hidden by the adversary, that is, values $y \in [M]$ satisfying $|f^{-1}(y)| \leq k - 1$, for a given parameter $k - 1 > 0$ known/chosen to/by the user. Additionally, the constructed QS also allows to decode sets $f^{-1}(y)$ for any $(k - 1)$-rare value of function $f$.

The following polynomial-time deterministic algorithm produces a universal QS solving $(N, M, \ell, k - 1)$-HFO, which has $O((k + \ell)k\ell \, \mathrm{polylog}(N, M, \ell, k))$ queries, as shown later in Theorem 1. Our construction algorithm uses constants $c_k, c_\ell \in (4, 8]$, which have to be chosen to satisfy

---

[2] In this paper we use a common asymptotic notation $O(f), \Omega(f), \Theta(f)$ to denote that a considered formula is, respectively, asymptotically upper bounded, lower bounded or equivalent to the function $f$ used inside the parenthesis, up to a constant factor.

conditions defined in the algorithm and in the analysis, but they do not influence asymptotic length of the constructed QS.[3] It constructs a 0-1 matrix $\mathcal{M}^*$ of $N \cdot M$ rows (each row corresponds to some pair $(x, y) \in [N] \times [M]$), which columns are transformed into queries at the very end of the construction. The algorithm exploits, among others, various aspects of polynomials over finite fields of size equal to (different) prime numbers.

**Main ideas of constructing queries and decoding feedback.** For each pair $(x, y)$, where $y$ is a $(k - 1)$-rare value and $x$ is in its preimage, we would like to have a corresponding query $Q_{z'}$ (i.e., column $z'$ in the constructed matrix) such that:

(1) there is 1 in the row labeled $(x, y)$ and column $z'$, and

(2) all other valid pairs $(x', y')$, where $(x', y') \neq (x, y)$ and $y'$ is $(k - 1)$-rare value and $x'$ is in the preimage of $y'$, have 0 in row $(x', y')$ and column $z'$.

This property would allow the decoding algorithm, executed by the user, to filter out all pairs $(x'', y'')$ having 1 in some column $z'$ receiving feedback 0. All pairs that remain are expected to be in the solution of the $(N, M, \ell, k - 1)$-HFOproblem.

To guarantee the properties (1) and (2) stated above, we select two sets of polynomials, take their cartesian products and evaluate over certain bounded-size field. Selecting properly polynomials and sizes of fields, we obtain that each pair $(x, y)$ in the problem solution has an argument $z$ on which the evaluations of the two associated polynomials are unique, in the sense that no other pair $(x', y')$ in the problem solution has the same pair when evaluated for argument $z$.

Now, we have to turn these pairs into some binary sequences, such that the binary blocks replacing column $z$ satisfy properties (1) and (2). This can be done by turning the evaluation of polynomials into prime numbers of the order corresponding to the evaluations, multiply the primes, and put 1 in the position corresponding to that multiplication result. This position will be unique for pair $(x, y)$, among other pairs $(x', y')$ in the problem solution, which completes the desription of main ideas.

**High-level description of the construction algorithm.** The construction algorithm proceeds in five steps, as described in detail in Figure 1. In Step 1, it sets up background parameters. In Step 2, a matrix $\mathcal{M}_{[N]}$ of $N$ rows is constructed, with values defined by different polynomials $P_x$, $x \in [N]$, evaluated over the elements of some bounded-size field (the field is over one of the prime number parameters defined in Step 1). Intuitively, some evaluation properties of polynomials $P_x$ will help to ensure the selection of elements in the inverse images of $(k-1)$-rare values in the final construction. Analogously, in Step 3, a matrix $\mathcal{M}_{[M]}$ of $M$ rows is constructed, with values defined by different polynomials $R_y$, $y \in [M]$, evaluated over the elements of some bounded-size field (over another prime parameter defined in Step 1). Intuitively, some evaluation properties of polynomials $R_y$ will assure selection of $(k-1)$-rare values out of other (at most $\ell - 1$) values of function $f$ in the final construction.

In Step 4, matrices $\mathcal{M}_{[N]}$ and $\mathcal{M}_{[M]}$ are combined into one matrix $\mathcal{M}$ of $N \cdot M$ rows and the number of columns equal to the sum of the numbers of columns in matrices $\mathcal{M}_{[N]}$ and $\mathcal{M}_{[M]}$. The entries in $\mathcal{M}$ are pairs of numbers, in which the first coordinate stores some corresponding number from $\mathcal{M}_{[N]}$ and the second coordinate stores some corresponding number from $\mathcal{M}_{[M]}$. In this way, each entry stores numbers coming from evaluation of two polynomials, say $P_x$ and $R_y$, from different sets of polynomials: one of them (the second coordinate of the entry) helps to guess a $(k - 1)$-rare value while avoiding guesses of any other $\ell - 1$ values of $f$, and the other one (the first coordinate of the entry) helps to identify "one by one" the inverse image of the selected $(k - 1)$-rare value. Informally speaking, by "helps to guess" we mean that in some column $z$ the considered pair of numbers (evaluations of two polynomials) will be unique among other pairs $(P_{x'}(z), R_{y'}(z))$ in the same column $z$ associated with some $(k - 1)$-rare value $y'$ and one of its counter images $x'$, where $(x', y') \neq (x, y)$. This will help the decoding algorithm to identify pair $(x, y)$. But before that, one more construction step is needed - making the matrix binary in a way to preserve the uniqueness described above.

More precisely, in Step 5, each entry $\mathcal{M}(x, y, z)$ of matrix $\mathcal{M}$, where $(x, y)$ is the row label and $z$ is the column label, is replaced by a specific 0-1 sequence $\sigma_{x,y,z}$. The positions of value 1 in

---

[3]The exact values of these constants can be deducted from the proof of Theorem 1.

**Input:** $N, M, \ell, k - 1$.

**Step 1:** *Setting parameters.* Let $d_k = \lceil \log_k N \rceil$, and let $q_k = c_k \cdot k d_k$ be a prime number such that $q_k^{d_k+1} = (c_k \cdot k d_k)^{d_k+1} \geq N$, for some constant $4 < c_k \leq 8$. Observe that such absolute constant $c_k$ exists, because $(k d_k)^{d_k+1} > N$ and between two integers: $\lceil (4 k d_k)^{d_k+1} \rceil$ and its double $2 \lceil (4 k d_k)^{d_k+1} \rceil \leq \lceil (8 k d_k)^{d_k+1} \rceil$, there is always at least one prime number.

We also define $d_\ell$ and $q_\ell$, by analogy: Let $d_\ell = \lceil \log_\ell M \rceil$, and let $q_\ell = c_\ell \cdot \ell d_\ell$ be a prime number such that $q_\ell^{d_\ell+1} = (c_\ell \cdot \ell d_\ell)^{d_\ell+1} \geq M$, for some constant $4 < c_\ell \leq 8$. Observe again that such absolute constant $c_\ell$ exists, because $(\ell d_\ell)^{d_\ell+1} > M$ and between two integers: $\lceil (4 \ell d_\ell)^{d_\ell+1} \rceil$ and its double $2 \lceil (4 \ell d_\ell)^{d_\ell+1} \rceil \leq \lceil (8 \ell d_\ell)^{d_\ell+1} \rceil$, there is always at least one prime number.

**Step 2:** *Defining matrix $\mathcal{M}_{[N]}$.* Consider polynomials $P_x$ of degree $d_k$ over field $[q_k]$, for $1 \leq x \leq q_k^{d_k+1}$. There are $q_k^{d_k+1}$ of such different polynomials. The role of this set of polynomials, $P_x$, in the final construction will be to ensure the selection of elements in the inverse images of $(k-1)$-rare values.

A matrix $\mathcal{M}_{[N]}$ of size $q_k^{d_k+1} \times q_k$ is constructed as follows. Each row $x$ contains subsequent values $P_x(z)$ of polynomial $P_x$ for arguments $z = 0, 1, \ldots, q_k - 1$, where $z$ is the column number. Then, matrix $\mathcal{M}_{[N]}$ is trimmed to $N$ rows by removing the excess $q_k^{d_k+1} - N$ rows (recall that $q_k^{d_k+1} \geq N$ by definition).

**Step 3:** *Defining matrix $\mathcal{M}_{[M]}$.* Analogously to Step 2, consider polynomials $R_y$ of degree $d_\ell$ over field $[q_\ell]$, for $1 \leq y \leq q_\ell^{d_\ell+1}$. There are $q_\ell^{d_\ell+1}$ of such different polynomials. The role of this set of polynomials, $R_y$, in the final construction will be to assure selection of $(k-1)$-rare values out of other (at most $\ell - 1$) values of function $f$.

Matrix $\mathcal{M}_{[M]}$ of size $q_\ell^{d_\ell+1} \times q_\ell$ is created analogously to $\mathcal{M}_{[N]}$ from Step 2, but using polynomials $R_y$ instead of $P_x$. Each row $y \in [M]$ contains subsequent values $R_y(z)$ of polynomial $R_y$ for arguments $z = 0, 1, \ldots, q_\ell - 1$, where $z$ is the column number. Then, matrix $\mathcal{M}_{[M]}$ is trimmed to $M$ rows by removing excess $q_\ell^{d_\ell+1} - M$ rows (recall that $q_\ell^{d_\ell+1} \geq M$ by definition).

**Step 4:** *Combining matrices $\mathcal{M}_{[N]}$ and $\mathcal{M}_{[M]}$ into matrix $\mathcal{M}$.* Matrix $\mathcal{M}$ of $N \cdot M$ rows and $q$ columns, where $q = q_k + q_\ell$, is constructed from $\mathcal{M}_{[N]}$ and $\mathcal{M}_{[M]}$ as follows. In each row labeled $(x, y) \in [N] \times [M]$, we take the corresponding polynomials $P_x, R_y$, and then for each column $z \in [q]$, we set $\mathcal{M}(x, y, z) = (P_x(z \mod q_k), R_y(z \mod q_\ell))$, where $(x, y)$ is the row label and $z$ is the column label.

**Step 5:** *Enhancing matrix $\mathcal{M}$ to get final $\mathcal{M}^*$.* The following procedure enhances matrix $\mathcal{M}$ to get a new (and final) matrix $\mathcal{M}^*$. Let $q_k', q_\ell'$ be the prime numbers of order $2q_k$ and $2q_\ell + 1$, respectively (i.e., the $(2q_k)$-th prime number in the order of all prime numbers, and $(2q_\ell + 1)$-th prime number in the order of all prime numbers, respectively), where the ordering starts from order 0 (and the corresponding prime 2). For each entry $\mathcal{M}(x, y, z) = (P_x(z \mod q_k), R_y(z \mod q_\ell))$, we proceed as follows. Let $x^* = P_x(z \mod q_k)$ and $y^* = R_y(z \mod q_\ell)$. Let $p_{x^\star}$ be the prime number of order $2x^\star$, and let $p_{y^\star}$ be the prime number of order $2y^\star + 1$. We create a 0-1 sequence $\sigma_{x,y,z}$ of length $q_k' \cdot q_\ell'$, by putting value 1 in column $p_{x^\star} \cdot p_{y^\star}$, and value 0 elsewhere. We call $\sigma_{x,y,z}$ a *segment $z$ of row $(x, y)$*. Then we replace the entry $\mathcal{M}(x, y, z)$ by the 0-1 sequence $\sigma_{x,y,z}$. After doing it for every entry of the original matrix $\mathcal{M}$, we obtain a 0-1 matrix that we call $\mathcal{M}^*$. The new matrix has the same number of rows, $N \cdot M$, as the previous one $\mathcal{M}$, while the number of columns is equal to $q \cdot (q_k' \cdot q_\ell')$.

**Output:** For every $i \leq q \cdot (q_k' \cdot q_\ell')$, query $Q_i$ is defined based on column $i$ of matrix $\mathcal{M}^*$ as follows: a pair $(x, y) \in [N] \times [M]$ is in $Q_i$ if and only if $\mathcal{M}^*(x, y, i) = 1$.

Figure 1: Algorithm constructing a universal QS, solving $(N, M, \ell, k - 1)$-HFO for *any hidden (by the adversary)* function $f : [N] \to [M]$.

**Input:** Matrix $\mathcal{M}^*$ of the universal QS (as constructed in Figure 1), solving $(N, M, \ell, k-1)$-HFO, for *any hidden (by the adversary)* function $f : [N] \to [M]$. Feedback vector $\Phi \in \{0, 1\}^{q \cdot q'_k \cdot q'_\ell}$.

**Initial:** Let $V = [N] \cdot [M]$ and $V^* = \emptyset$.

**Decode:** For any row label $(x, y)$ of matrix $\mathcal{M}^*$: If there is a position $z$ such that $\mathcal{M}^*(x, y, z) = 1$ and $\Phi(z) = 0$ then remove $(x, y)$ from $V$.

**Verify:** For every $y$ such that there is at least one pair $(x, y) \in V$: if the number of pairs in $V$ with the second coordinate equal to $y$ is at most $k - 1$, put all pairs in $\{(x, y) : (x, y) \in V\}$ to set $V^*$.

**Output:** Set $V^*$.

Figure 2: Decoding algorithm for the constructed universal QS (from Figure 1) solving $(N, M, \ell, k-1)$-HFO, for *any hidden (by the adversary)* function $f : [N] \to [M]$.

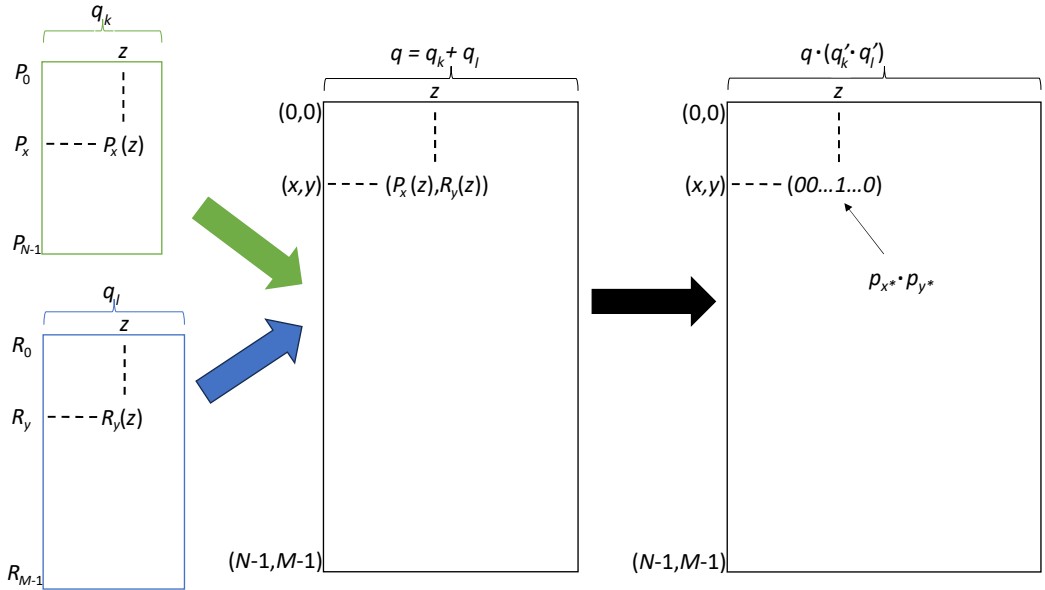

Figure 3: Two matrices on the left are $\mathcal{M}_{[N]}$ and $\mathcal{M}_{[M]}$, respectively at the top and bottom. They are constructed is Steps 2 and 3, resp., of the construction algorithm in Figure 1 of the main paper. The matrix in the middle is $\mathcal{M}$, constructed from $\mathcal{M}_{[N]}$ and $\mathcal{M}_{[N]}$ in Step 4. The argument $z$ of $P_x$ is modulo $q_k$ and the argument $z$ of $R_y$ is modulo $q_\ell$. The matrix on the right is the final matrix $\mathcal{M}^*$, which replaces each entry of each column $z$ of $\mathcal{M}$ by a $z$-segment, which is a 0-1 sequence of length $q'_k \cdot q'_\ell$ with only one value 1 at position $p_{x^*} \cdot p_{y^*}$ defined in Step 5 of the construction algorithm in Figure 1 of the main paper.

these sequences depend on the values of polynomials $P_x(z)$ and $R_y(z)$, turned into primes – this is to guarantee certain distributions of "single occurrences of 1's" that, together with the uniqueness property described in Step 4, allow successful decoding based on the binary OR feedback.

For the sake of convenience in description of Step 5 and its analysis, w.l.o.g. we assume that prime numbers are ordered starting from 0, i.e., prime of order 0 is 2, of order 1 is 3, of order 2 is 5, etc.

The resulting 0-1 matrix $\mathcal{M}^*$ is then, as mentioned earlier, transformed, column by column, into a query system.

**High-level description of the decoding algorithm.** The decoding algorithm is presented in Figure 2. It takes the matrix $\mathcal{M}^*$, constructed in Figure 1, as part of the input. The other part of the input is the 0-1 feedback $\Phi$ obtained from the adversary after applying the QS defined by $\mathcal{M}^*$ to

the function $f$ (chosen&hidden by the adversary). It filters out all pairs $(x', y')$ that are inconsistent with feedback $\Phi$, and returns set $V^*$ of some remaining pairs for which, after the initial filtering, the second coordinate (a candidate for a $(k-1)$-rare value) occurs in at most $k-1$ pairs.

More detailed arguments are presented in the proof of Theorem 1 that follows.

**Theorem 1.** *The query system constructed by the algorithm in Figure 1, $\{Q_i\}_{i=1}^{q \cdot (q'_k \cdot q'_\ell)}$, together with the associated decoding algorithm in Figure 2, solve the $(N, M, \ell, k-1)$-HFO problem using $q \cdot (q'_k \cdot q'_\ell) = (k+\ell)k\ell \cdot \text{polylog}(N, M, \ell, k)$ queries, where $\text{polylog}(N, M, \ell, k) = O\left( \frac{\log(NM)}{\log \min\{k,\ell\}} \cdot \frac{\log N}{\log k} \cdot \frac{\log M}{\log \ell} \cdot (\log k + \log\log N) \cdot (\log \ell + \log\log M) \right)$, for some absolute constant parameters $4 < c_k, c_\ell \leq 8$ of the construction hidden in the Big O asymptotic notation. The construction itself and the algorithm decoding $(k-1)$-rare values with their inverse images are polynomial in $N, M$.*

*Proof.* Consider a constructed matrix $\mathcal{M}^*$ for some suitable absolute constants $c_k, c_\ell \in (4, 8]$. Observe first that two polynomials $P_x$ and $P_{x'}$ of degree at most $d_k$, for $i \neq j$, could have equal values for at most $d_k$ different arguments. This is because they have equal values for arguments $z$ for which $P_x(z) - P_{x'}(z) = 0$. However, since $P_x - P_{x'}$ is also a polynomial of degree at most $d_k$, it could have at most $d_k$ zeroes. Hence, $P_x(z) = P_{x'}(z)$ for at most $d_k$ different arguments $z$. Analogous argument applies to polynomials $R_y$ and their upper bound of the degree, $d_\ell$.

**Useful notation.** Let us fix a function $f$ with the image containing at most $\ell$ values. Consider a $(k-1)$-rare value $y$ of function $f$ (if it does not exist, the proof concludes immediately) and any set $K$ such that $K \supset f^{-1}(y)$ and $|K| \leq k$. Pick an arbitrary argument $x \in K$. Consider the row labeled $(x, y)$ in $\mathcal{M}^*$. Let $P_x$ be the corresponding polynomial based on the first coordinate of this row's label, and $R_y$ be the polynomial corresponding to the second coordinate $y$. Let $\mathcal{P}$ be the set of all polynomials corresponding to the rows with the first coordinate in $K$ and second equal to $y$, and let $\mathcal{P}' = \mathcal{P} \setminus \{P_x\}$. Consider the image subset $L = f([N]) \setminus \{y\} \subseteq [M] \setminus \{y\}$; by assumption on function $f$, we get $|L| \leq \ell - 1$. Define $\mathcal{R} = \{R_{y'} : y' \in L\}$.

**Formulating "uniqueness" properties.** We first argue that there is a segment $z \in [q]$ such that

    (a) $P_x(z) \neq P_{x'}(z)$ for any $x' \in K \setminus \{x\}$, and

    (b) $R_y(z) \neq R_{y'}(z)$, for every $y' \in L$.

**Proof of the "uniqueness" properties for some $z$.** Consider the following Condition1$_x(z)$ and observe that Event1$_x$ holds:

> *Condition1$_x(z)$*: some of the at most $k-1$ polynomials in $\mathcal{P}'$ could be equal to $P_x$ on argument $z$, modulo $q_k$.
> *Event1$_x$*: There are at most $(k-1) \cdot d_k$ different arguments $z \in [q_k]$ satisfying Condition1$_x(z)$.

Note that in fact we only consider different $z \in [q_k]$, as polynomials $P$ are defined over the field $[q_k]$. Hence, the upper bound on the number of arguments $z \in [q]$ satisfying Condition1$_x(z)$ is $(k-1)d_k \cdot \left\lceil \frac{q}{q_k} \right\rceil$.

By analogy, consider the following Condition2$_y(z)$ and observe that Event2$_y$ holds:

> *Condition2$_y(z)$*: some of the at most $\ell-1$ polynomials in $\mathcal{R}$ could be equal to $R_y$ on argument $z$, modulo $q_\ell$.
> *Event2$_y$*: There are at most $(\ell-1) \cdot d_\ell$ different arguments $z \in [q_\ell]$ satisfying Condition2$_y(z)$.

Again, here we only consider different $z \in [q_\ell]$, as polynomials $R$ are defined over the field of arguments in $[q_\ell]$. Hence, the number of arguments $z \in [q]$ satisfying Condition2$_y(z)$ is at most $(\ell-1) \cdot d_\ell \cdot \left\lceil \frac{q}{q_\ell} \right\rceil$.

To summarize the two previous paragraphs, none of Condition1$_x$ and Condition2$_y$ happens for at least the following number of arguments $z \in [q]$:

$$
\begin{aligned}
q - (k-1)d_k \cdot \left\lceil \frac{q}{q_k} \right\rceil - (\ell-1) \cdot d_\ell \cdot \left\lceil \frac{q}{q_\ell} \right\rceil \;\; &> \;\; q - kd_k \cdot \left( \frac{q}{q_k} + 1 \right) - \ell \cdot d_\ell \cdot \left( \frac{q}{q_\ell} + 1 \right) \\
&\geq \;\; q - \frac{q}{c_k} - \frac{q_k}{c_k} - \frac{q}{c_\ell} - \frac{q_\ell}{c_\ell} \\
&\geq \;\; q - \frac{3q}{\min\{c_k, c_\ell\}} \\
&\geq \;\; 1 \,,
\end{aligned}
\tag{1}
$$

since $q \geq q_k + q_\ell$ and $c_k, c_\ell > 4$.

Consider such an argument $z$ and its corresponding segments in the constructed matrix $\mathcal{M}^*$. Consider column $p_{x^\star} \cdot p_{y^\star}$ in this segment, where $p_{x^\star}$ is the prime numbers of order $2P_x(z \mod q_k)$, and $p_{y^\star}$ is the prime number of order $2R_y(z) + 1$. By definition, this column of segments $z$ has 1 in row $(x, y)$, but not in any other rows $(x', y) \in (K \setminus \{i\}) \times \{y\}$, by the fact that $z$ does not satisfy Condition$1_x(z)$; hence, property (a) holds. This column does not have a 1 in rows with second coordinate in set $L$, as such rows are not divisible by $p_{y^\star}$, because $z$ does not satisfy Condition$2_y(z)$; hence, property (b) holds. This completes the proof of the "uniqueness" properties for some $z$. We use this important property in the correctness argument below.

**Concluding correctness analysis – from the "uniqueness" properties to successful decoding.** To finalize the correctness argument, first observe that for every $(k-1)$-rare value $y$ of function $f$, all pairs in $f^{-1}(y) \times \{y\}$ successfully pass the **Decode** part of the decoding algorithm (Figure 2), as they contribute to the feedback by the OR operator. They could, however, hypothetically do not be put to set $V^*$ in the **Verify** part of the decoding algorithm. This is possible only if at least one pair $(x, y)$, for some $x \notin f^{-1}(y)$, had passed the **Decode** part as well.

Suppose, to the contrary, that it had happened. Consider set $K = f^{-1}(y) \cup \{x\}$. It has at most $k - 1$ elements. Therefore, we proved already that there is a column $z$ such that $\mathcal{M}^*(x, y, z) = 1$ while $\mathcal{M}^*(x', y, z) = 0$ for every $x' \in f^{-1}(y)$ and $\mathcal{M}^*(x'', y', z) = 0$ for any $x''$ and any $y' \neq y$. This however implies that $\Phi(z) = 0$, which means that $(x, y)$ would not have passed the **Decode** part of the decoding algorithm. This is a contradiction, concluding the proof that the constructed QS correctly solves the $(N, M, \ell, k-1)$-HFO problem for the user.

**Number of queries.** The length of the constructed QS is $q \cdot (q'_k \cdot q'_\ell) = O((q_k + q_\ell)q_k q_\ell \log q_k \log q_\ell)$, as $q'_k = O(q_k \log q_k)$ and $q'_\ell = O(q_\ell \log q_\ell)$ by the prime number theorem applied to the largest of the considered prime numbers – the prime numbers of order $2q_k$ and $2q_\ell + 1$, resp. We have $q_k = O(k \log N / \log k)$ and $q_\ell = O(\ell \log M / \log \ell)$. Consequently, $\log q_k = O(\log k + \log \log N)$ and $\log q_\ell = O(\log \ell + \log \log M)$. Hence, the length of the constructed QS is

$$
q \cdot (q'_k \cdot q'_\ell) \;\; = \;\; O((q_k + q_\ell)q_k q_\ell \log q_k \log q_\ell) \;\; \leq \;\; (k + \ell)k\ell \cdot \text{polylog}(N, M, \ell, k) \,,
$$

where the $\text{polylog}(N, M, \ell, k)$ is upper bounded by

$$
O\left( \frac{\log(NM)}{\log \min\{k, \ell\}} \cdot \frac{\log N}{\log k} \cdot \frac{\log M}{\log \ell} \cdot (\log k + \log \log N) \cdot (\log \ell + \log \log M) \right) \,.
$$

**Time complexity analysis.** The construction is clearly polynomial, as the number of considered polynomials $P, R$ is at most $N \cdot M$, the number of columns is also polynomial in $N, M$, and computing values of these polynomials and using sieve methods (or other efficient algorithms) to compute prime numbers of the considered orders are also polynomial. Similarly, the decoding algorithm is also polynomial, as it scans at most $M + N$ times through each row (there are $N \cdot M$ rows) and compares it with the feedback vector $\Phi$ in polynomial time. $\qquad\square$

## 2.1 Enhancing the Universal QS for HFO by Fault-tolerant Guarantees

Suppose one would like to be able to decode the hidden set correctly even if some $\alpha$ positions in the feedback vector would be altered by a *worst-case* adversary. More precisely, assume that the adversary could change the feedback vector in some $\alpha$ positions of its choice. Observe that if we use larger constants $c_k, c_\ell$ in the construction, for instance, $c_k > 2 + \frac{\alpha}{kd_k}$ and $c_\ell > 2 + \frac{\alpha}{\ell d_\ell}$, there will be always some column with correct feedback. Specifically, in the proofs of Theorem 1, the number

of segments (and thus, also columns) for a pair $(x, y)$ to have different values in this segment than other pairs $(x', y)$, satisfying $f(x') = y$, and pairs $(x'', y')$, satisfying $y' \neq y$, is at least (see Eq. (1))

$$
\begin{aligned}
q - (k-1)d_k - (\ell-1)d_\ell &= (c_k k d_k + c_\ell \ell d_\ell) - (k-1)d_k - (\ell-1)d_\ell \\
&> \left(2 + \left\lceil \frac{\alpha}{k d_k} \right\rceil\right) k d_k + \left(2 + \left\lceil \frac{\alpha}{\ell d_\ell} \right\rceil\right) \ell d_\ell - (k-1)d_k - (\ell-1)d_\ell \,,
\end{aligned}
$$

which subtracted by the number of adversarially changed ones, $\alpha$, is still at least 1 (as required in Eq. (1)). Enhancing constant $c_k, c_\ell$ increases the lengths of selectors by factor at most $\left\lceil \frac{\alpha}{k d_k} + \frac{\alpha}{\ell d_\ell} \right\rceil$. Decreasing the number of queries with respect to the number of tolerated faults or generalizing to other types of adversarial/stochastic failures is an interesting open direction.

## 3 LOWER BOUND ON THE LENGTH OF ANY UNIVERSAL QS FOR HFO

Below we show a lower bound that nearly matches Thm. 1.

**Theorem 2.** *Every universal QS system solving* $(N, M, \ell, k - 1)$*-HFO has the number of queries*

$$
\Omega(\ell \cdot \min\{k^2 \log_k N, N\}) \,.
$$

*Proof.* Consider a universal QS system $\mathcal{Q}$ solving $(N, M, \ell, k - 1)$-HFO for any hidden function $f$ bounded by these parameters. Our goal is to specify one of such functions, which enforces the number of queries in this QS to be in $\Omega(\ell \cdot \min\{k^2 \log_k N, N\})$.

Consider a potential set of values $L \subseteq [M]$ of size $\ell$, of such function $f$. Let us fix any $y \in L$, and consider a set of pairs $[N] \times \{y\} = \{(x, y) : x \in [N]\}$. We argue that the number of queries in the QS system $\mathcal{Q}$ that

- contain at least one pair in $[N] \times \{y\}$, and

- do not have any pair in $[N] \times (L \setminus \{y\})$

is at least $\Omega(\min\{k^2 \log_k N, N\})$. We denote the set of these queries by $C_y$, where by definition $C_y \subseteq \mathcal{Q}$. Indeed, suppose that the number of such queries is smaller than $c \cdot \min\{k^2 \log_k N, N\}$, for any arbitrary constant $c > 0$. It follows that these queries cannot form an $(N, k)$-superimposed code, by the lower bound $\Omega(\min\{k^2 \log_k N, N\})$ on the length of any $(N, k)$-superimposed code, c.f., Clementi et al. (2001).

(Here, by "queries forming a superimposed code" we understand that the elements of the set $[N] \times \{y\}$ correspond 1-1 to codewords, and codeword labeled $(x, y)$ has 1 in position $j$ if the $j$th considered query contains the pair $(x, y) \in [N] \times \{y\}$. A code is an $(N, k)$-superimposed code if for any set $K$ of at most $k$ codewords there is a codeword $v \in K$ which is not contained in a Boolean OR of the codewords in $K \setminus v$.)

It further implies that there is a subset $K$ of $[N] \times \{y\}$ of size at most $k$ and an element $(x, y) \in K$ such that for any considered query $Q$ in $C_y$, if $(x, y) \in Q$ then $Q \cap (K \setminus \{(x, y)\}) \neq \emptyset$. In other words, sets $K$ and $K \setminus \{(x, y)\}$ return the same feedback by the considered queries in $C_y$. If we define specific functions $f_1, f_2$ such that $f_1^{-1}(y) = \{x' : (x', y) \in K\}$ and $f_2^{-1}(y) = \{x' \neq x : (x', y) \in K\}$, then our QS returns the same inverse image set of $k$-rare value $y$ for both $f_1, f_2$, as it is based on the same feedback obtained for sets $f_1^{-1}(y) \times \{y\}$ and $f_2^{-1}(y) \times \{y\}$ (as argued earlier). This is a contradiction, because these two returned sets should be different, by definition of $f_1^{-1}(y), f_2^{-1}(y)$. This concludes the proof of the lower bound $\Omega(\min\{k^2 \log_k N, N\})$ on the considered set of queries $C_y$.

Next, we consider a family of sets $\mathcal{C} = \{C_{y'} : y' \in L\}$, where each set $C_{y'}$ of queries is defined analogously to the above generic definition of $C_y$. Any $C_y, C_{y'}$ in this family $\mathcal{C}$, where $y \neq y'$, are disjoint, by the second bullet in the generic definition of $C_y$ above. There are $\ell = |L|$ sets $C_y$ in the considered family $\mathcal{C}$. Hence, the total number of queries in the considered QS system $Q$ should be at least

$$
\sum_{y \in L} |C_y| = |L| \cdot \Omega(\min\{k^2 \log_k N, N\}) \geq \Omega(\ell \cdot \min\{k^2 \log_k N, N\}) \,.
$$

$\square$

**Additional Discussion, Open Problems and Limitations in Appendix A and B.**

**Reproducibility Statement.** (This optional reproducibility statement is not part of the main text and therefore will not count toward the page limit.) This paper is focused on theoretical problems – the problem and all required notation are defined formally in the main paper. Construction and decoding algorithms are stated in descriptive pseudo-codes formats in Figures 1 and 2, respectively. They are formally analyzed (correctness and performance) in Theorem 1. The claimed nearly-matching lower bound is formally proved in Theorem 2. All assumptions are clearly stated in definitions, and potential limitations are discussed in Appendix B.

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

# APPENDIX

## A  DISCUSSION AND OPEN DIRECTIONS

This work introduced new concepts of learning rare values of hidden functions using substantially smaller number of queries then could follow from the established area of group testing. We designed polynomially constructable (and decodable) nearly optimal, in terms of the number of queries, non-adaptive deterministic query systems that learn rare values and their inverse images of any hidden function in time polynomial in function sparsity $\ell$ (i.e., the number of values in the function image, instead of the whole codomain) and in threshold $k$ defining rare values. To justify near-optimality of our construction, we also proved the corresponding lower bound.

The construction and decoding algorithms could be represented by an OR-NOT logic circuit. The input are all possible pairs of potential arguments and values. The first internal layer contains nodes corresponding to the constructed queues, being OR-gates connected to the included pairs; this layer computes feedback vector. The second layer contains NOT logic gates, one per each OR gate from layer one – this is to reverse logic values of the feedback vector. Then, the output layer is again the set of all pairs of potential arguments and values, $(x, y)$, each being an OR logic gate, connected to those NOT gates in layer two which already have a path to the input node corresponding to the same pair $(x, y)$. This is to guarantee filtering out pairs that are not in the problem solution. Finally, the output pairs with value $0$, i.e., not filtered out by OR-gate operation on reversed feedback in layer two, are taken to the problem solution. (Alternatively, we could once again apply another NOT layer to reverse the filtering outcome, after which nodes/pairs with value $1$ are taken to the problem solution.) This circuit is polynomially constructed, based on the algorithms in Figures 1 and 2 and Theorem 1, it has a constant depth and the number of internal gates is asymptotically the same as the number of queries stated in Theorem 1 (in particular, it depends on $N, M$ only polylogarithmically). One may also ask if the polynomial dependence of the number of queries (and the size of the corresponding logic circuit) on parameters $k, \ell$ is practical or not? [4] The outliers parameter $k$ is (close) to constants in applications, by the meaning of outliers (rare elements). The size of the image could be larger, and therefore practitioners should take this aspect into account and seek other solutions for large values of $\ell$.

Below we discuss interesting open directions emerging from this work.

**Complexity gaps.**  The most straightforward future direction is to improve the remaining polylogarithmic gaps on the lengths of the query systems solving HFO.

**Multi-dimensional and structured systems.**  Another promising future direction is to study whether more structured domains/codomains, e.g., multi-dimensional or metric, (hyper-)graphs or matroids, etc. could improve the number of queries even further. If so, what properties of domain/codomain make substantial impact to query complexity?

**Other feedback functions.**  Extending our construction and lower bound to other types of feedback function, considered in the literature (cf., Klonowski et al. (2022)) is another interesting research direction. Different feedback, such as quantitative or parity feedbacks, can in some cases sub-

---

[4]The dependence is quadratic in $k, \ell$ when considered separately, and cubic when considered jointly.

stantially (e.g., nearly quadratically) improve the number of queries, c.f., Bshouty (2009); Censor-Hillel et al. (2015).

**Adversarial settings.** There are many adversarial models that could be analyzed. In case of false positives changes to the feedback: even in case of a malicious Byzantine adversary, surprisingly, if the number of false positives feedback changes is bounded by $\alpha = O(\min\{kd_k, \ell d_\ell\})$, our construction can tolerate them if we adjust constants $c_k, c_\ell$ with factors $\alpha/(kd_k), \alpha/(\ell d_\ell)$, respectively. After such adjustment, our construction guarantees more than $\alpha$ "good" occurrences of every important pair of elements (rare value and one of its inverse images) in the feedback (i.e., occurrences that avoid other similar pairs or pairs with other values). Therefore, even if the adversary introduces $\alpha$ false-positive changes to the feedback, our QS recognizes them and rejects. It automatically extends to more benign adversaries with smaller adaptivity. If one wants to tolerate more than $O(k\log_k n)$ twists in the feedback without asymptotic increase on the number of queries in the QS, the problem remains open. We hypothesize that it could work for less adaptive adversaries, but for the Byzantine adversary an impossibility result is more likely.

**Use of randomization.** Randomization may possibly help in discovering rare values of a hidden function, especially against an oblivious adversary who has to decide about the function in advance. An interesting twist would be to consider a dynamic adaptive adversary who may dynamically "tailor" the hidden function, as long as it is compatible with the feedback obtained so far. If randomness helps against such adversary, a new question arises about the minimum amount of randomness (entropy) needed and how it affects the number of queries?

# B  LIMITATIONS

**Non-adaptive (universal) QS.** There is a distinction between an *adaptive* QS, which allows for designing subsequent queries one-by-one, each time applying the knowledge of the results of the preceding queries, and a *non-adaptive* QS, which requires that all queries are designed in advance, without any information about results of other queries. This work considers only non-adaptive query systems. While adaptive approach might, under some assumptions, result in more efficient QS schemes – i.e., with a smaller number of queries – there are some advantages of non-adaptive query systems which make it attractive. One of them is parallelization. While adaptive QS schemes can be executed only in a *sequential* way (because each consecutive query may depend on the results of the previous queries), all the queries of the non-adaptive scheme can be executed in *parallel*. This property may, e.g., give much faster testing scenarios. Another advantage of non-adaptive QS could be better resiliency to failures, which we proved is indeed the case in the considered problem.

**Simple single-threshold feedback with $s = 1$.** We focus on a popular and simple variant of a very general single-threshold feedback model with threshold $s = 1$ (OR-queries), but there are other popular feedbacks considered in the literature. cf., Klonowski et al. (2022).

**Known parameters.** We assume that all parameters $(N, M, \ell, k, \alpha)$, except the function $f$ itself, are known in advance to the constructing and decoding algorithms. Parameters $N, M$ are necessary parts of the game, to be shared by both players (the user and the adversary). Parameters $k, \alpha$ could be chosen by the user, depending on the level of "rareness" of elements and fault-tolerance the user wants to be guaranteed. Parameter $\ell \leq M$ could be hidden from the user, who may instead use a classic doubling technique to estimate $\ell$ when building the QS (and decoding), starting from low values.

