# OpenReview forum: "Testing for Outliers in a Hidden Function"
_ICLR.cc/2026/Conference — ICLR 2026 Conference Withdrawn Submission_

### Official Review · Reviewer_h6vb · 2025-11-02

**Soundness:** 3
**Presentation:** 3
**Contribution:** 2
**Rating:** 6
**Confidence:** 3

**Summary:**

This paper studies outlier detection and formulates it by introducing the Hidden Function Outlier (HFO) problem. This is a new theoretical model for identifying rare outputs of an unknown function f:[N]→[M] using only non-adaptive OR-queries over input–output pairs. The authors define as “outliers” or “k-rare” those among the $\ell$ points in the range of the function $f$ whose preimage has size at most $k$.
This framework generalizes classical group testing to functional settings, enabling the discovery of all such “outlier” values y.
More specifically, the authors design a deterministic, polynomial-time constructible and decodable query system that requires only $(k+\ell)k\ell\cdot\mathrm{polylog}(N,M,\ell,k)$ queries, which is nearly cubic in the sparsity and rarity parameters, but only polylogarithmic in the domain and range sizes. Each query is a set of input-output pairs for the function $f$ and the “feedback” that the user receives from each query is binary. That is, the feedback is “1” if at least one of the pairs is correct and “0” otherwise.
 A matching lower bound shows this dependence is essentially optimal up to logarithmic factors.
The construction relies on finite-field polynomials and prime encodings to ensure each rare pair $(x,y)$ produces a unique query signature, allowing reliable reconstruction even under limited adversarial noise.

**Strengths:**

The paper’s primary strength lies in its conceptual originality, as it introduces the Hidden Function Outlier (HFO) problem, extending classical group testing to a more general and functional setting. The authors provide nearly tight upper and lower bounds on query complexity, along with a deterministic, polynomial-time constructible and decodable query system.

**Weaknesses:**

As the authors acknowledge in the appendix, the framework presented is restricted to non-adaptive query systems, which, while easier to parallelize, may require more queries than adaptive ones. It also relies on a binary feedback model (OR-queries), limiting its generality compared to having more refined feedback. The approach also assumes full prior knowledge of the relevant parameters, which might be unrealistic in many real-world scenarios.

**Questions:**

- Do you believe that adding adaptivity to the queries in the HFO problem would yield asymptotically fewer queries, or do you expect similar lower bounds to hold?
-How difficult would it be to extend the construction to other feedback types (e.g., asking for the fraction of queried input-output pairs that are correct)? Would it help to have this additional information?

---

### Official Review · Reviewer_Gh4w · 2025-11-03

**Soundness:** 3
**Presentation:** 3
**Contribution:** 2
**Rating:** 2
**Confidence:** 3

**Summary:**

The  paper studies the problem of finding outliers in a function that is given through a query access and presents an efficient (in terms of queries) algorithm to find the set of all outliers. More formally, given access to an unknown function and an oracle that given as input a set of pairs {x_i , y_i } where x_i is in the domain and y_i is in the codomain, outputs where any of these points "fits" the function. Given such access the aim of the algorithm is to find the set of all points that are images of the function at most $k$ times. The paper presents an algorithm for this problem that runs in linear time in $k$ and image size of $f$ and logarithmically on the domain and codomain sizes. This can be seen as closely related to the problem of combinatorial group testing.

**Strengths:**

The problem studied by the paper I think could be interesting in many settings simlar to the applications of combinatorial group testing. Algorithm presented also seems like an interesting extension of algorithmic ideas in the CGT area.

**Weaknesses:**

As presented, the model comes off as a bit arbitrary. In addition, the current presentation does not motivate the algorithm and is presented abruptly. In addition, I don't immediately see the topic of the paper as being highly relavent to the ICLR audience.

**Questions:**

-> Similar to CGT, is there a simple way in which the algorithm can be abstracted in terms of error correcting codes? and the construct can be seen as a transformation of the Reed-Solomon code? This would help the reader further appreciate the construction.
-> I think it would help motivate the algorithm (especially for readers not familiar with CGT) if the problem is rephrased in terms of binary matrices briefly. That would help reader see where the algorithm comes from
-> If indeed the authors think there are interesting (potential/theoretical) applications in machine learning, it would help to discuss this further. The current presentation does not seem address at the ICLR audience

---

### Official Review · Reviewer_Vud5 · 2025-11-05

**Soundness:** 4
**Presentation:** 3
**Contribution:** 2
**Rating:** 4
**Confidence:** 2

**Summary:**

The paper is in a broad sense motivated by outlier detection.
It studies a clean theoretical setup:
- two players A and B play a game, where B knows some function $f: X\to Y$ such that $|f(X)|\leq l$,
- $A$ knows only $X$, $Y$ and $l$
- a value $y\in Y$ is $k$-rare if $|f^{-1}(y)|\leq k$
- the goal of A is to find all the $k$-rare values and their preimages
- A is allowed to ask B queries, where each query is a subset $Q\subset X\times Y$. The answer is whether or not $Q$ contains any pair $(x, y)$ s.t. $y=f(x)$.

The goal of the paper is to develop an algorithm to create a small sequence of non-adaptive queries that A can use. Here non-adaptivity means that the sequence queries must be provided up-front instead of being created interactively as answers are revealed.

The paper provides some, I believe, non-trivial upper and lower bounds on the size of such a sequence of queries.

**Strengths:**

- clear theoretical setup
- main argument is mostly easy to follow on a high level

**Weaknesses:**

- very limited relevance to the broader ML community; currently, this is more of a CS theory / combinatorics paper.

**Questions:**

- it would be great if you could elaborate on the relevance of this problem to machine learning by further modeling and/or empirical work

---

### Official Review · Reviewer_sY2E · 2025-11-05

**Soundness:** 2
**Presentation:** 3
**Contribution:** 3
**Rating:** 2
**Confidence:** 3

**Summary:**

The paper introduces the Hidden Function Outlier (HFO) problem: given an unknown function $f:[N]\to[M]$ with image size at most $\ell$, identify all k-rare values (values with preimage size $\leq k$) using non-adaptive OR-queries. The authors propose a deterministic universal query system of length $\tilde O((k+\ell)k\ell)$ (polylogarithmic in N,M), a decoding algorithm, and a lower bound of $\Omega(\ell \cdot \min\{k^2\log_k N, N\}$). They also sketch a fault-tolerant extension and argue relevance to anomaly detection and group testing.

**Strengths:**

- The paper presents a clear formalization of the HFO problem and its parameters.
- Deterministic construction with correctness proof and polynomial-time decoding.
- Provides a lower bound that connects to superimposed codes, giving some theoretical grounding.
- Attempts to address fault tolerance, albeit superficially.

**Weaknesses:**

- Query complexity is cubic in $k,\ell$ and requires an $N\times M$ matrix, making it infeasible for large domains.
- The claimed tightness seems overstated.  Upper and lower bounds differ significantly when $\ell \gg k$; “nearly matching” claim is misleading.
- Fault-tolerance analysis is weak. It simply inflates constants without rigorous trade-off or adversarial model.

**Questions:**

Given the strengths, I have the following concerns:

- The paper argues that naïve GT on $[N]\!\times\![M]$ is inefficient, but does not compare against two‑stage/product constructions or Cartesian‑product cover‑free families that might reduce dependence on $N, M$ while isolating the image values first (e.g., by first finding $\ell$ image elements and then decoding preimages). Without such comparisons, it’s unclear how much the proposed design actually improves over straightforward alternatives grounded in the same toolbox.

-  How does your construction fundamentally differ from known explicit non-adaptive group testing schemes (e.g., [1])? Section 1.3 does not do justice. Could you formalize the gap?

-  The query complexity is cubic in $(k,\ell)$ up to polylog factors. In many realistic “outlier” regimes k may be small, but \ell (the image size) can be moderate/large, making $(k+\ell)k\ell$ potentially large. Moreover, the construction builds an incidence matrix with $N\cdot M$ rows and very long blocks (length $q'_k q'_\ell$​), which undercuts practical scalability. Can the authors clarify?

- In addition, I believe that the fault‑tolerance “extension” increases constants rather than fundamentally changing asymptotics. Specifically, the fault‑tolerance section is heuristic: it inflates constants $c_k,c_\ell$​ to guarantee “at least one good column,” but gives no tight trade‑off or rigorous adversary model beyond a simple bound on the number of flips. Can the authors provide a rigorous analysis of error resilience (e.g., minimum distance, decoding under $\alpha$ adversarial flips) rather than heuristic constant inflation?

-  For $\ell \gg k$, the upper bound is $\tilde O(\ell^2 k)$ while the lower bound is $\Omega(\ell k^2\log_k N)$. Can you clarify why this is “nearly matching” and whether improvements are possible?


- From the practicability persepective, what is the actual size of $M^\*$ for realistic $N,M,k,\ell$? Could compression or sparsity-aware techniques make this feasible?


[1] Porat & Rothschild. Explicit nonadaptive combinatorial group testing schemes

---

### Note · Authors · 2025-11-21

I have read and agree with the venue's withdrawal policy on behalf of myself and my co-authors.